# Establishing Height-for-Age Z-Score Growth Reference Curves and Stunting Prevalence in Children and Adolescents in Pakistan

**DOI:** 10.3390/ijerph191912630

**Published:** 2022-10-03

**Authors:** Muhammad Asif, Muhammad Aslam, Iqra Mazhar, Hamza Ali, Tariq Ismail, Piotr Matłosz, Justyna Wyszyńska

**Affiliations:** 1Department of Statistics, Government Associate College, Qadir Pur Raan 60000, Pakistan; 2Department of Statistics, Bahauddin Zakariya University, Multan 60000, Pakistan; 3Department of Preventive Pediatrics, The Children Hospital and Institute of Child Health, Multan 60000, Pakistan; 4Department of Biology, Government Associate College, Qadir Pur Raan 60000, Pakistan; 5Department of Food Sciences and Technology, Bahauddin Zakariya University, Multan 60000, Pakistan; 6Institute of Physical Culture Sciences, Medical College, University of Rzeszów, 35-959 Rzeszów, Poland; 7Institute of Health Sciences, Medical College, Rzeszów University, 35-959 Rzeszów, Poland

**Keywords:** children, growth curves, height, LMS method, Pakistan, stunting

## Abstract

Height-for-age Z-score (HAZ) curves are widely used for detecting children with stunting. The aim of this study was to provide smoothed HAZ growth reference values and their curves for Pakistani children and adolescents aged 2–18 years. The prevalence of stunting in Pakistani children was determined. A total of 10,668 healthy Pakistani participants were included. Information related to age, sex, city of residence and height (cm) was used. Age- and sex-specific smoothed HAZ growth reference values and associated graphs were obtained using the lambda-mu-sigma (LMS) method. The prevalence of stunting was calculated by applying WHO 2007 and USCDC 2000 height-for-age references and local reference of the study population. In both sexes, the smoothed HAZ curves increased with age. For 2 and 3 years of age, the height values of the girls were greater than those of the boys. The boys were then taller than the girls. Comparing our median height (z = 0) growth reference data from WHO, USCDC and corresponding data from other countries, Pakistani children and adolescents have significantly different reference values than their counterparts in the reference group. A marked overestimation of the prevalence of stunting was observed (10.8% and 17.9% according to WHO and USCDC, respectively) compared to the local reference (3.0%). It is recommended that the prevalence of stunting in children and adolescents is determined by applying local height growth references to plan health strategies and treatments in the local population.

## 1. Introduction

Malnutrition is a severe and life-threatening disease that affects more than 2 billion people worldwide [1]. Stunting is a kind of malnutrition, and the World Health Organization (WHO) considers a child too small for his/her age when the height-for-age Z-score (HAZ) value is less than negative two standard deviations (i.e., <−2SD) from the median height of the WHO reference population [2]. The most common symptoms of stunting are decreased physical and cognitive development, which can have long-term negative consequences [3]. Stunting in childhood age can also lead to decreased bone density, decreased productivity, delayed maturation, later brain development, poor school performance, impaired immune systems, poorer cognitive development, deficiencies in muscular strength and reduced work capacity [4,5,6]. Past studies on factors associated with stunting indicated that maternal and father education, increasing child age, increasing birth order, lack of sanitation facilities, duration of breastfeeding, presence of diarrhea and wealth index were associated with stunting [7,8].

The World Bank Group, UNICEF and WHO have recently estimated that there are more than 151 million children with stunting under the age of five throughout the world, with South Asian nations having the highest prevalence (almost 57.9 million) [9,10]. Although the rate of stunting generally decreased in Asia from 38% to 23% between 2000 and 2017, it is still considered the highest percentage [11]. According to the 2018 Global Nutrition Report, Pakistan has the third highest number of children with stunting in the world (10.7 million), after Nigeria (13.9 million) and India (46.6 million) [10]. Local estimates based on data from the 2017 Pakistan National Nutritional Survey indicate that the Gilgit Baltistan region has the highest stunting percentage (46%), followed by the provinces of Sindh (45.5%), Khyber Pakhtunkhwa (KPK) (40%), Punjab (36.4%) and Kashmir (24.2%) [12].

For the determination of infants and children with stunting, height-for-age growth standards or growth references are of particular importance. In 2006, the WHO provided child growth standards [13] and growth references [14] for school-aged children and adolescents in 2007, and the U.S. Centers for Disease Control and Prevention (USCDC) [15] released growth references in 2000. References from the WHO and the USCDC have been widely used to detect children with stunting; however, due to genetic and environmental differences, these references may not be applicable to other populations, particularly in developing countries [16]. In 2014, Natale and Rajagopalan conducted a systematic review in which they compared the WHO growth reference data with data from 55 countries and concluded that due to the wide variations in anthropometric measurements, the WHO reference is not appropriate for all countries and could put children at risk of misdiagnosis [17]. In Iran, Payandeh et al. [18] also compared height-for-age curves with WHO standard curves and concluded that WHO growth standards may not properly recognize populations at risk of growth problems in Iranian children. Similarly, many other countries have also shown that WHO or USCDC reference charts are not applicable for their populations [19,20,21,22].

Several countries [23,24,25] have produced height-for-age growth reference curves using national sample populations. Few studies have been published in Pakistan that report age-related height growth reference charts for school-aged children. For example, Mushtaq et al. [26] used the LMS approach to present height-for-age growth references for children aged 5 to 12 years. A more recent study by Qaisar and Karim [27] created height-for-age growth charts only for Pakistani girls aged 8 to 16 years. However, HAZ growth reference values and their curves for children and adolescents aged 2 to 18 years are scarce and are currently required. We constructed the missing charts in response to this requirement.

The primary objective of our study was to provide smoothed HAZ growth reference values and their curves for Pakistani children and adolescents aged 2–18 years. We also calculated the prevalence of stunting in children and adolescents using growth references from local studies, as well as WHO and USCDC growth references.

## 2. Materials and Methods

The study used a secondary type of dataset collected in the 2016 Multiethnic Anthropometric Survey (MEAS). Details about the study design, sampling strategy and inclusion/exclusion criteria for survey respondents have been extensively described elsewhere [20,21,22,28,29]. Briefly, in this survey, 10,782 children and adolescents aged 2 to 19 years were enrolled in Pakistan four main cities (Lahore, Multan, Rawalpindi and Islamabad). The dataset was compiled from 68 public and private schools and included 9929 children and adolescents aged 5 to 19 years. The complete list of schools in the respective cities was taken from the Punjab Department of Education (schools). The lists of schools in respective cities were stratified according to school grade (primary, secondary and higher secondary), and probability sampling was used for school selection. Participants were sampled from each class based on simple random sampling. Data on subjects younger than five years old were acquired in public places, including markets, shopping centers and parks, using convenient sampling. For this study, we included 10,668 children and adolescents aged 2 to 18 years to make the HAZ curves using the LMS statistical method. The authors assert that this study followed the complete study protocol declared by Helsinki [30].

In this survey, information related to sociodemographic variables was included, such as age (years), sex status (boys/girls) and city of residence (Multan/Lahore/Rawalpindi/Islamabad), along with different anthropometric measurements, i.e., body weight, height, waist circumference (WC), hip circumference (HpC), neck circumference (NC), head circumference (HdC) and mid-upper arm circumference (MUAC). All anthropometric measurements were taken under a standard procedure. Complete measurement details have been discussed in previously published studies [20,21,22,28]. A Seca 217 stadiometer was used for height measurement, and all circumference measurements of each subject were taken using non-elastic plastic tape, standing in a standard procedure as described in studies [20,21,22,28].

For the descriptive analysis, frequencies (n) along with percentages (%) and mean ± standard deviation (SD) were estimated for both boys and girls, varying at different ages. Mean height differences between two groups were determined using an unpaired *t*-test, and analysis of variance (ANOVA) was used for mean comparisons of more than two groups. The HAZ growth reference values for both boys and girls were obtained using the LMS method [31]. The height-for-age percentile values (C) were also calculated using the equation *C* = *M* [(1 + *LSZ*)^1/*L*^], where *Z* is the *Z*-score of the normalized distribution [31]. For example, for an 18-year-old boy with smoothed L = 1.48, M = 165.19 and S = 0.033 along with SD score (Z = 1.645), the value of the 95th percentile was obtained as C_95_ = 165.19 [{1 + (−1.48) × (0.033) × (1.645)} ^1/−1.48)^] = 178.26. Similarly, we obtained the 95th percentile value for girls using three smoothed parameters L ≤ 0.17, M = 156.43 and S = 0.040, along with Z = 1.645 as C_95_ = 156.43 [{1 + (−0.17) × (0.040) × (1.64)} ^1/(−0.17)^] = 167.06. These values are provided in (Appendix A). For the detection of children and adolescents with stunting, WHO growth references [13,14], USCDC 2000 references [15] and local growth references obtained in this study were used. For example, according to WHO growth references, boys 6 years of age who have height < -2SD values or < 116.0 cm are classified as children with stunting. Using USCDC 2000 growth references, height < 5th percentile values or < 107.31 cm indicates stunting in the same demographic. Cohen’s κ statistic was used to assess the agreement on the classification of stunting between international references USCDC, WHO and our study population references. The values of κ < 0.6 and κ ≥ 0.90 were considered to be in poor and excellent agreement, respectively [32]. A *p*-value < 0.05 was considered statistically significant in the whole analysis. The study was approved by the Departmental Ethics Committee of Bahauddin Zakariya University, Multan, Pakistan (IRB# SOC/D/2715/19). The software Statistical Package for Social Sciences (SPSS) version 21.0 and the R language were used for all statistical analyses.

## 3. Results

A total of 10,668 subjects (boys = 5539 (51.9%) and girls = 5129 (48.1%)) were enrolled in this study. The mean (± SD) age and height of the total subjects was 10.54 (±3.96) years and 137.67 (±19.55) cm, respectively. In both sexes, mean height increased with age. The age, sex, and area-wise mean comparison showed that the mean height of boys was greater than that of girls. For overall subjects, the mean height in boys was also significantly higher than in girls (boys vs. girls: 142.31 ± 19.51 cm vs. 132.66 ± 10.33 cm; *p* < 0.05). The average height of boys belonging to different ethnicities was not statistically significant, F(2, 5536) = 1.05, *p* = 0.349, while in girls of different ethnicities, average height values were significantly different, i.e., F(2, 5126) = 30.97, *p* < 0.001 (Table 1).

The sex-specific HAZ growth reference values are shown in Table 2 and their smoothed curves are shown in Figure 1 (A+B). Height growth reference values increased with age in both sexes. For 2 and 3 years of age, the height values of the girls for z = ±1, ±2 and ±3 were greater than those of the boys. The boys were then taller than the girls. Overall, the total increase in the median height (z = 0) was 72.3 cm for boys and 60.5 cm for girls. Between 2 and 16 years of age, an annual increase in median height was approximately 4 to 7 cm for both sexes. The sex-specific comparison of our median height growth reference data (z = 0) with data from WHO [13,14], USCDC [15] and of other countries [23,24,25,26] is presented in Table 3. The figures of children and adolescents with stunting were also calculated. The local study reference indicated a comparatively low prevalence of stunting in various age groups compared to the USCDC and WHO references. In the overall sample, 3.0% of the subjects (3.4% boys and 2.5% girls) were stunted using references from the local study. Using WHO and USCDC references, the prevalence of stunting was 10.8% (8.8% boys and 12.8% girls) and 17.9% (16.0% boys and 19.9% girls), respectively. The prevalence of stunting among subjects aged 11–19 was lower (2.9%) compared to subjects aged 2–10 years (3.1%). Boys were more prone to stunting in both age groups as well as in different ethnicities than girls (Table 1). Cohen’s kappa statistic values indicate that there is poor agreement between the local references and the USCDC and WHO references (κ = 0.080, 0.147, respectively).

## 4. Discussion

To the best of our knowledge, this study presents the HAZ growth reference data for the first time for a wide age range (from 2 to 18 years) of children and adolescents in Pakistan. The present study also reports the age-, sex- and ethnicity-specific prevalence of stunting using WHO, USCDC and local study growth references. In this study, we analyzed the data of 10,668 children and adolescents aged 2 to 18 years obtained in the MEAS-2016 survey. These survey data have already been used in different studies for constructing the reference charts of body mass index (BMI), WC and NC [20,21,22]. The aforementioned studies indicated that boys had significantly (*p* < 0.05) larger mean values of BMI, NC and WC than girls in various age groups. Similar to other anthropometric characteristics, the results of our study indicate that boys had higher mean height values than girls. Our HAZ growth reference curves show that height increased with age in both sexes, which is consistent with earlier published research with Chinese [23], Indian [24] and Turkish [25] children. A study of Indian children and adolescents aged 3–18 years indicated that boys had higher median height reference values than girls [24]. Other studies [23,25] with full sets of age range (i.e., 2–18 years) also presented higher mean height reference values for boys than girls. Our research results are parallel to these findings.

Various studies [25,26,27] explained a significant difference in height growth references between countries and between different ethnic groups of the same country; for example, studies with Chinese [23], Indian [24] and Pakistani children [26] found that the median height data were very different from the median data of the WHO and the USCDC at all ages. Consistent with these reports, our median height reference values (z = 0) for younger children (i.e., 2–7 years of age) were higher than the WHO and USCDC growth references. Thus, both boys and girls were shorter not only than WHO and USCDC references, but also shorter than Chinese [23], Indian [24] and Turkish children [25]. The disparity in results is probably due to genetic and environmental differences, and the disparity infers that the use of WHO and USCDC growth references as standards for Pakistani children would be seriously misleading. If epidemiological researchers and pediatricians use these standards, they can overestimate the frequency of stunting in children. This statement was also proved in a recent Pakistani study by Qaisar and Karim [27], who found that the general prevalence of stunting among girls was markedly higher when using USCDC and WHO references compared to local references. A kappa correlation indicated poor agreement between the local references and the USCDC and WHO references in that study. Parallel to previous study findings [27], our study also indicates a significantly higher prevalence of stunting in children and adolescents using WHO and USCDC growth references, along with poor agreement.

In local studies over the past two decades, the prevalence of stunting was observed to decrease among Pakistani schoolchildren. For example, the Pakistan National Health Survey (NHSP 1990-94) reported that the prevalence of stunting among children is 16.7% (boys = 16.8% and girls = 16.6%), which decreased to 14.6% (boys = 15.5% and girls = 13.8%) in the Karachi Health Survey 2004–2005 and to 8% (boys = 8.6% and girls = 7.7%) in the 2011 Lahore-based nutritional assessment survey [33,34]. Parallel to these, the overall prevalence of stunting of 3.0% (boys = 3.4% and girls = 2.5%) using local references in our study also showed a remarkable improvement in the nutritional status of children. The sex-wise comparison indicated that boys had a higher prevalence of stunting than girls, which was also consistent with the results of earlier surveys [33,34]. Using WHO and USCDC references, we found that children 9 years of age or older were more prone to stunting and the prevalence was higher among girls than among boys. This finding is in line with the study conducted in four districts of Sindh, Pakistan [35]. Another study conducted with girls aged 8–16 years from the most populated Punjab province, Pakistan, found an increasing trend of stunting with increasing age when using USCDC and WHO references [27].

The current study and local published studies did not include height-for-age growth reference data of newborn children or children up to 2 years of age, and the results of this study suggest that this gap might be filled because a sufficient increase in height is shown during this age. Furthermore, socioeconomic status also has a considerable influence on the body size of children [34,35]; therefore, comparative studies should also be planned that include socioeconomic data.

## 5. Conclusions

Using the LMS method, HAZ growth reference data based on a comprehensive sample of Pakistani children aged 2 to 18 years were presented. The analysis showed a significant disparity between our growth references and references of the WHO, USCSC and of other international population references. A marked difference in the estimation of stunting prevalence among children 2–18 years old after applying WHO, USCDC and local height references indicates that each country should use its own growth reference charts since standard charts used from other countries provide misleading information. Population-specific height references can be used to determine and monitor stunting in children and adolescents. More research is also required to overcome the limitations of the present study.

## Figures and Tables

**Figure 1 ijerph-19-12630-f001:**
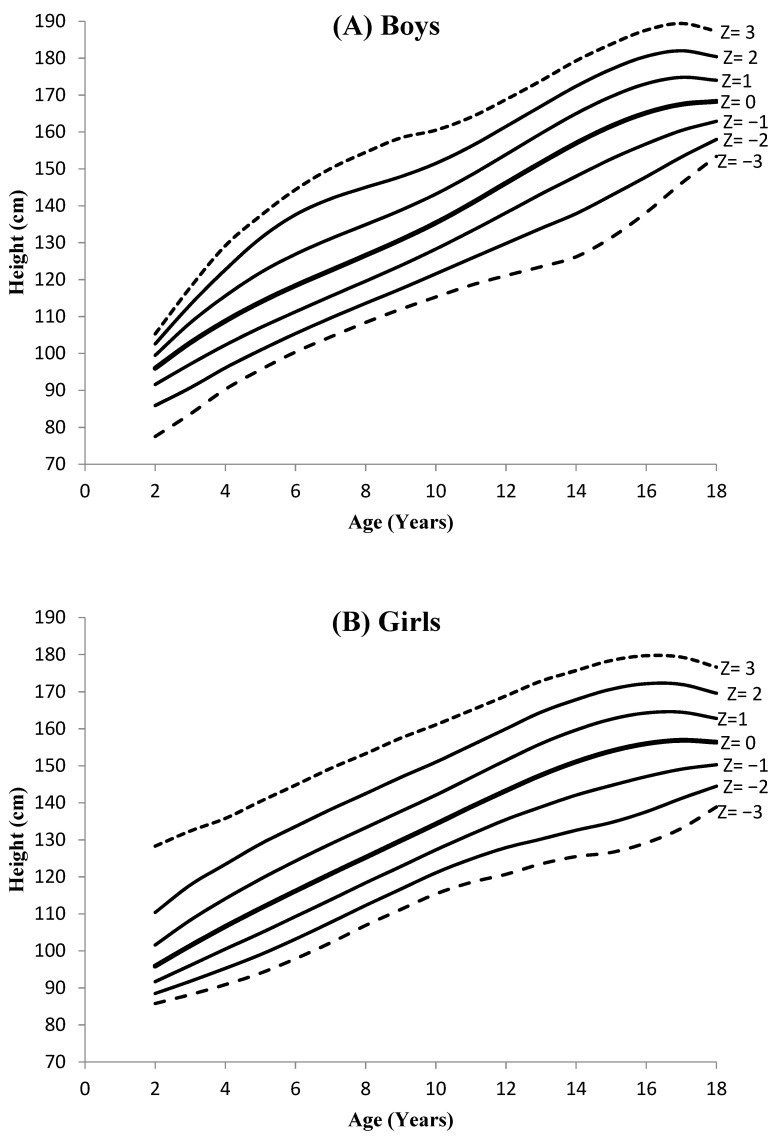
Height-for-age *Z*-score growth reference curves using the LMS method for Pakistani children and adolescents, aged 2–18 years.

**Table 1 ijerph-19-12630-t001:** Descriptive statistics of the participants by age, sex and city of residence.

Age (Years)	Boys					Girls				
	N	Mean ± SD	Stunting ^a^ (%)	Stunting ^b^ (%)	Stunting ^c^ (%)	N	Mean ± SD	Stunting ^a^ (%)	Stunting ^b^ (%)	Stunting ^c^ (%)
02	19	96.58 ± 4.03	5.3	0	0	43	96.37 ± 4.75 ^NS^	0	0	0
03	100	102.69 ± 5.82	1.0	0	0	170	101.71 ± 6.50 ^NS^	1.8	0	0
04	208	108.54 ± 8.43	8.2	2.4	2.9	313	106.45 ± 7.91 ^S^	5.4	2.9	2.2
05	240	112.78 ± 7.13	5.8	5.0	5.0	381	111.30 ± 7.38 ^S^	3.4	4.2	6.6
06	292	119.36 ± 7.81	3.4	4.5	7.9	405	117.15 ± 6.99 ^S^	1.2	2.0	4.0
07	279	123.70 ± 7.18	2.5	3.9	6.1	381	122.21 ± 7.64 ^S^	1.6	3.1	8.4
08	273	128.21 ± 7.63	2.9	2.6	7.3	376	127.55 ± 8.07 ^NS^	1.6	2.9	8.0
09	247	132.65 ± 7.18	3.2	3.6	8.9	336	130.73 ± 6.65 ^S^	1.2	5.7	11.3
10	420	139.19 ± 7.76	2.1	3.3	14.5	459	132.59 ± 7.93 ^S^	5.2	15.9	26.4
11	439	140.09 ± 7.89	2.3	5.9	15.0	325	138.37 ± 8.05 ^NS^	1.2	17.2	21.8
12	675	144.45 ± 8.28	2.4	9.3	19.1	436	142.97 ± 8.60 ^S^	3.9	26.8	34.4
13	593	149.07 ± 9.28	5.2	18.0	25.5	460	148.96 ± 8.77 ^NS^	1.5	23.7	38.3
14	563	156.52 ± 9.28	5.2	16.0	23.8	341	150.50 ± 8.87 ^S^	2.6	30.8	38.7
15	546	162.98 ± 7.77	1.6	11.2	17.6	257	153.33 ± 8.27 ^S^	1.2	23.3	42.8
16	381	165.83 ± 7.37	1.8	11.3	21.0	191	155.97 ± 9.81 ^S^	3.1	19.4	30.4
17	169	167.66 ± 6.41	3.0	7.7	23.1	129	157.01 ± 5.74 ^S^	0	10.1	20.9
18	95	167.78 ± 6.71	7.4	16.8	32.6	126	157.33 ± 5.74 ^S^	4.8	11.1	21.4
*Age groups (years)*										
2–10	2078	123.23 ± 12.92	3.6	3.4	7.7	2864	120.16 ± 12.76 ^S^	2.7	5.2	9.4
11–18	3461	153.77 ± 12.61	3.3	12.1	21.0	2265	148.47 ± 10.37 ^S^	2.3	22.6	33.2
02–18	5539	142.31 ± 19.51	3.4	8.8	16.0	5129	132.66 ± 10.33 ^S^	2.5	12.8	19.9
*City of residence*										
Lahore	2839	142.20 ± 19.30	2.7	8.3	16.0	2091	135.04 ± 19.52 ^S^	2.1	11.5	18.7
R. pindi/Isl	1764	142.81 ± 21.01	2.8	7.5	13.7	1926	131.35 ± 17.01 ^S^	2.6	14.0	21.0
Multan	936	141.73 ± 18.01	6.7	12.9	20.6	1112	130.46 ± 17.59 ^S^	3.1	13.3	20.3
		F = 1.05_(*p* = 0.349)_					F = 30.97_(*p* = 0.000)_			

^a^ Stunting prevalence using local study reference. ^b^ Stunting prevalence using WHO reference. ^c^ Stunting prevalence using USCDC reference. SD: standard deviation; R. pindi: Rawalpindi; Isl: Islamabad; ^S^: significant *p* < 0.05; ^NS^: not significant.

**Table 2 ijerph-19-12630-t002:** Height-for-age Z-score growth reference values for Pakistani children and adolescents, aged 2–18 years.

Age (Years)	L	S	M	Z = −3	Z = −2	Z = −1	Z = 0	Z = 1	Z = 2	Z = 3
**Boys**										
2	5.73	0.041	95.95	77.5	85.9	91.6	96.0	99.5	102.6	105.3
3	2.51	0.054	102.92	83.6	90.7	97.1	102.9	108.3	113.2	117.9
4	0.23	0.061	108.81	90.3	96.1	102.3	108.8	115.6	122.7	129.2
5	−1.23	0.065	113.88	95.6	100.9	107.0	113.9	121.9	131.2	137.3
6	−1.98	0.065	118.36	100.4	105.4	111.3	118.4	126.9	137.6	144.4
7	−2.15	0.063	122.51	104.5	109.6	115.5	122.5	131.1	141.9	150.1
8	−1.85	0.060	126.57	108.4	113.6	119.6	126.6	134.9	145.0	154.5
9	−1.20	0.057	130.80	112.0	117.5	123.8	130.8	138.8	147.9	158.4
10	−0.32	0.055	135.44	115.3	121.6	128.3	135.4	143.2	151.5	160.5
11	0.68	0.054	140.64	118.5	125.7	133.1	140.6	148.3	156.1	164.0
12	1.64	0.054	146.18	121.1	129.8	138.1	146.2	153.9	161.5	168.8
13	2.43	0.054	151.73	123.5	133.9	143.2	151.7	159.6	167.0	173.9
14	2.88	0.054	156.97	126.2	137.9	148.0	157.0	165.0	172.4	179.3
15	2.87	0.052	161.56	131.4	142.8	152.7	161.6	169.6	177.0	183.8
16	2.24	0.049	165.19	138.2	147.9	156.8	165.2	173.1	180.5	187.6
17	0.84	0.043	167.53	146.1	153.2	160.4	167.5	174.8	182.0	189.4
18	−1.48	0.033	168.25	153.4	158.0	162.9	168.3	174.0	180.4	187.3
**Girls**										
2	−5.20	0.050	95.91	85.8	88.5	91.7	95.9	101.6	110.4	128.3
3	−3.31	0.059	101.41	88.2	91.8	96.1	101.4	108.3	117.8	132.4
4	−2.13	0.063	106.59	90.1	95.3	100.5	106.6	114.1	123.4	135.8
5	−1.50	0.065	111.51	94.0	99.0	104.8	111.5	119.4	128.9	140.4
6	−1.26	0.064	116.23	97.9	103.2	109.3	116.2	124.3	133.6	144.8
7	−1.25	0.062	120.82	102.2	107.7	113.8	120.8	128.9	138.2	149.3
8	−1.33	0.059	125.32	106.9	112.3	118.4	125.3	133.3	142.5	153.3
9	−1.32	0.057	129.79	111.2	116.7	122.8	129.8	137.7	146.9	157.5
10	−1.08	0.055	134.31	115.4	121.1	127.3	134.3	142.1	151.0	161.1
11	−0.50	0.055	138.87	118.5	124.8	131.5	138.9	146.8	155.5	165.0
12	0.29	0.056	143.33	120.7	127.9	135.5	143.3	151.5	160.0	168.9
13	1.13	0.058	147.47	123.5	130.2	138.9	147.5	156.0	164.5	172.9
14	1.84	0.058	151.10	125.5	132.6	142.1	151.1	159.7	167.9	175.7
15	2.23	0.058	154.01	126.6	134.7	144.7	154.0	162.6	170.7	178.4
16	2.15	0.055	156.01	129.2	137.6	147.1	156.0	164.3	172.2	179.7
17	1.40	0.049	156.88	133.1	141.2	149.1	156.9	164.5	172.0	179.3
18	−0.17	0.040	156.43	138.9	144.5	150.3	156.4	162.8	169.6	176.6

**Table 3 ijerph-19-12630-t003:** Comparison of the height (cm) median growth reference values (Z = 0) for Pakistani children and adolescents with WHO, USCDC and other international studies.

Age (Years)	Present Study	WHO [13,14]	USCDC [15]	Mushtaq et al. (2012) [26]	Marwaha et al. (2011) [24]	Neyzi et al. (2015) [25]	Zong and Li (2013) [23]
**Boys**							
2	96.0	87.1	86.8			88.2	88.50
3	102.9	96.0	95.2		101.2	96.8	96.80
4	108.8	103.3	102.5		106.8	104.0	104.1
5	113.9	110.1	109.1	113.3	112.4	110.4	111.1
6	118.4	116.0	115.6	118.2	117.7	116.1	117.7
7	122.5	121.7	122.0	123.5	122.9	121.5	124.0
8	126.6	127.3	128.1	128.6	128.1	126.9	130.0
9	130.8	132.6	133.7	133.3	133.3	132.1	135.4
10	135.4	137.8	138.8	137.2	138.6	137.6	140.2
11	140.6	143.1	143.7	140.5	144.2	143.8	145.3
12	146.2	149.1	149.3	143.2	150.1	150.6	151.9
13	151.7	156.0	156.4		156.1	157.7	159.5
14	157.0	163.2	164.1		161.4	164.9	165.9
15	161.6	169.0	170.1		165.9	170.3	169.8
16	165.2	172.9	173.6		169.2	173.4	171.6
17	167.5	175.2	175.3		171.9	175.0	172.3
18	168.3	176.1	176.1		174.3	176.2	172.7
**Girls**							
2	95.9	85.7	85.4			86.80	87.20
3	101.4	95.0	94.2		99.4	95.40	95.60
4	106.6	102.7	101.0		105.2	102.5	103.1
5	111.5	109.6	107.9	114.1	110.8	109.1	110.2
6	116.2	115.1	115.0	118.7	116.4	115.1	116.6
7	120.8	120.8	121.7	123.4	121.9	121.1	122.5
8	125.3	126.6	127.8	128.2	127.6	126.7	128.5
9	129.8	132.5	133.1	133.3	133.5	132.1	134.1
10	134.3	138.6	138.2	138.5	139.4	137.9	140.1
11	138.9	145.0	144.2	143.7	145.0	145.4	146.6
12	143.3	151.2	151.4	148.6	149.8	153.1	152.4
13	147.5	156.4	157.3		153.3	157.8	156.3
14	151.1	159.8	160.4		155.5	160.4	158.6
15	154.0	161.7	161.9		156.9	161.7	159.8
16	156.0	162.5	161.4		157.6	162.4	160.1
17	156.9	162.9	162.5		158.1	162.7	160.3
18	156.4	163.1	162.9		158.5	163.1	160.6

## Data Availability

The data presented in this study are available on request from the corresponding author.

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
