# Peer review of "Establishing Height-for-Age Z-Score Growth Reference Curves and Stunting Prevalence in Children and Adolescents in Pakistan"

_ijerph, 2022, doi:10.3390/ijerph191912630_

Round 1
Reviewer 1 Report
[IJERPH] Manuscript ID: ijerph-1895453
General comments:
The authors address a very important topic. To their credit, they considered the WHO growth patterns as well as those posted by the US CDC when evaluating the observed z-scores among children and adolescents in Pakistan.
L44 - Introduction
This is a well-written section. Genetic variations and family history are always challenging; but a child’s growth tracking is important, regardless of %ile on the respective charts.
L47 - This issue of malnutrition is significant among Pakistani children. Yet, the authors only provide short, glib comments to this important point.
L77 – As the authors provide interesting HAZ data, what about other anthropometrics and their respective z-scores?
L81 - Materials and Methods
L82 – despite the use of 2016 data, were data on other anthropometrics available? If so, (L94), why were not these data discussed…must extend beyond height-for-age data.
L97 – never use “etc” in a scientific manuscript
L120 – Results
The other anthropometric data should be provided in this section. Such data should be presented as percentiles.
L132 - Table 1 – the significant figures are excessive.
L165 – The graphics are great; the HAZ data should be posted in a supplement file; the plotted data should be overlayed with respective data from WHO and CDC. Remember, most growth charts are posted as percentiles…not relative to z-scores.
L172 - Discussion
L173 – the authors’ work is not the first of its kind in Pakistan. See Aziz et al., 2012. Growth centile charts (Anthropometric measurement) of Pakistani pediatric population. J Pakistan Med Assn. 62(4):367-77)
This section is incomplete without the other anthropometric data from the nearly 11,000 subjects.
L227 – Conclusions
This section must consider the additional anthropometric data as mentioned on L93 and beyond.
The authors’ need to conduct a better review of the literature.
Author Response
Dear Reviewer,
Thank you for the opportunity to revise our manuscript, entitled “Establishing Height-for-age Z-score Growth Reference Curves and Stunting Prevalence in Children and Adolescents” [IJERPH-1895453]. We have included all changes as suggested by Reviewer. The article has been also proofread by a native speaker.
We hope that our revision meets your approval.
Thank you for your time and for your consideration of this manuscript, I am looking forward to hearing from you.
Sincerely Yours,
Authors
Following is the point-to-point response for your further necessary action.
Response to reviewer 1
Introduction
Point 1: This is a well-written section. Genetic variations and family history are always challenging; but a child’s growth tracking is important, regardless of %ile on the respective charts.
Response 1: Thank you for the comment. For more tracking on child’s growth, we have added height-for- age percentile values in Table S1 (supplementary table 1)
Point 2: This issue of malnutrition is significant among Pakistani children. Yet, the authors only provide short, glib comments to this important point.
Response 2: More detail has been provided in the revised manuscript.
Point 3: As the authors provide interesting HAZ data, what about other anthropometrics and their respective z-scores?
Response 3: Thank you for considering such an important point. In the materials and methods section, we have already described that the study used secondary data collected in the multi-ethnic anthropometric survey (MEAS) and the dataset was also publically available in the Mendeley (https://data.mendeley.com/datasets/sxgymx5xjm/1). In the literature, different studies are available that have already used the same MEAS data for constructing the growth reference charts of body mass index, neck circumference, head circumference, waist circumference, waist-to-height ratio. The complete details about the reference data of these anthropometrics can be seen in the following listed articles.
- Asif, M.; Aslam, M.; Altaf, S.; Mustafa, S. Developing waist circumference, waist-to-height ratio percentile curves for Pakistani children and adolescents aged 2–18 years using Lambda-Mu-Sigma (LMS) method. Pediatr. Endocrinol. Metab. 2020, 33, 983–993.
- Asif, M.; Aslam, M.; Khan, S.; Altaf, S.; Ahmad, S.; Qasim, M.; Ali, H.; Wyszyńska, J. Developing neck circumference growth reference charts for Pakistani children and adolescents using the lambda–mu–sigma and quantile regression method. Public Health Nutr. 2021, 24, 5641–5649.
- Asif, M., Aslam, M., Wyszyńska, J., Altaf, S. Establishing body mass index growth charts for Pakistani children and adolescents using the Lambda-Mu-Sigma (LMS) and quantile regression method. Minerva Pediatr,
- Asif M, Aslam M, Ismail T, Rahman A, Saleem N. Developing growth reference charts for the head circumference of Pakistani children aged 6 to 18 years. Turk J Pediatr. 2022; 64(2):293-301.
- Asif, M.; Aslam, M.; Qasim, M.; Altaf, S.; Ismail, A.; Ali, H. A dataset about anthropometric measurements of the Pakistani children and adolescents using a cross-sectional multi-ethnic anthropometric survey. Data Br. 2021, 34, 106642.
However, a complete set of height-for age z-score (HAZ) reference data for detecting children with stunting was not published up till now. We therefore, presented HAZ data in this study.
Materials and Methods
Point 4: despite the use of 2016 data, were data on other anthropometrics available? If so, (L94), why were not these data discussed…must extend beyond height-for-age data.
Response 4:
The details about the other anthropometric variables (beyond height) have been described in the revised manuscript.
Point 5: never use “etc” in a scientific manuscript
Response 5: Thank you for this comment! It has been removed.
Results
Point 6: The other anthropometric data should be provided in this section. Such data should be presented as percentiles.
Response 6: Already discussed, please refer to the response of point 3.
Point 7: Table 1 – the significant figures are excessive.
Response 7: The mean height values of boys were significantly greater than those of girls. The significant was tested at 5% in the entire analysis.
Point 8: The graphics are great; the HAZ data should be posted in a supplement file; the plotted data should be overlayed with respective data from WHO and CDC. Remember, most growth charts are posted as percentiles…not relative to z-scores.
Response 8: Very right. However, according to the WHO guidelines, a child will be stunted if he/she has the height-for-age z-score (HAZ) value less than minus two standard deviations (i.e., < -2SD) from the median height of WHO reference population. In this study, we also determined the prevalence of stunted children. Due to these findings, we presented the reference data in the form of z-scores. In addition, Z = -2 corresponds to the 2.3rd percentile values of the data set and Z = 0 indicates the median (P50) percentile values of the data set. We can present the reference data either in percentile or z-score form. Moreover, our study title also reflects the use of Z-scores.
Discussion
Point 9: the authors’ work is not the first of its kind in Pakistan. See Aziz et al., 2012. Growth centile charts (Anthropometric measurement) of Pakistani pediatric population. J Pakistan Med Assn. 62(4):367-77)
Response 9: Thank you so much for highlighting the published article. We want to explain here that the above mentioned study included the 12800 children aged 3-16 years of age from the four provinces of Pakistan. Even with available population datasets, the use of advanced statistical technique required to construct growth centile curves is compulsory. While they determined the reference values by using Epi-info software. They didn’t use the renowned LMS statistical method which has already been used by the WHO and different developed countries [23-26] for determining the reference values. International Obesity Task Force (IOTF) also has been adopted the same methodology in order to develop global BMI growth curves for children and adolescents. We can say that the reference values delivered by Aziz et al., 2012 cannot be used for determining stunting in children and adolescents at local level. In this study, we presented the height-for-age z-score growth reference values based on the sample including 2-18 years of aged Pakistani children and adolescents. These reference values would be suitable for Pakistani pediatrician to monitor their patient’s growth.
Point10: This section is incomplete without the other anthropometric data from the nearly 11,000 subjects.
Response 10: Revised with more detail.
Conclusions
Point 11: This section must consider the additional anthropometric data as mentioned on L93 and beyond.
Response 11: Revised with more detail.
Point 12: The authors’ need to conduct a better review of the literature.
Response 12: Thank you for this comment, review has been conducted.
Reviewer 2 Report
Dear authors,
First of all, I would like to thank you for the opportunity to review this interesting work. The study presented in this article is of sufficient interest and quality to be published in this prestigious journal. Nevertheless, I would like to make the following comments:
- Introduction: the introduction presented is correct, although possibly scarce. I would suggest that the state of the question should be examined in greater depth. In addition, and as a recommendation of the introduction and the article in general, there are very few references from the last 5 years. I suggest that at least the references from the last 5 years should be between 65 and 70% of the total number of references used.
- Materials and methods: I suggest making a new section that only includes information on the sample. This will make it easier for the reader to analyse the article. In reference to the contents, I think they are appropriate, I only suggest a change in the way it is presented. I also suggest that an explanation be given as to why this article was written with data taken in 2016, when there is a 6-year gap between the data collection and the analysis presented in this article.
- Results: they are presented clearly and systematically, complying with scientific requirements.
- Discussion: the discussion presented is correct, but again I suggest including more current references from the last 5 years.
Author Response
Dear Reviewer,
Thank you for the opportunity to revise our manuscript, entitled “Establishing Height-for-age Z-score Growth Reference Curves and Stunting Prevalence in Children and Adolescents” [IJERPH-1895453]. We have included all changes as suggested by Reviewer. The article has been also proofread by a native speaker.
We hope that our revision meets your approval.
Thank you for your time and for your consideration of this manuscript, I am looking forward to hearing from you.
Sincerely Yours,
Authors
Following is the point-to-point response for your further necessary action:
Dear authors,
First of all, I would like to thank you for the opportunity to review this interesting work. The study presented in this article is of sufficient interest and quality to be published in this prestigious journal. Nevertheless, I would like to make the following comments:
- Introduction: the introduction presented is correct, although possibly scarce. I would suggest that the state of the question should be examined in greater depth. In addition, and as a recommendation of the introduction and the article in general, there are very few references from the last 5 years. I suggest that at least the references from the last 5 years should be between 65 and 70% of the total number of references used.
Response: Thank you for this comment. The comment has been included.
- Materials and methods: I suggest making a new section that only includes information on the sample. This will make it easier for the reader to analyse the article. In reference to the contents, I think they are appropriate, I only suggest a change in the way it is presented. I also suggest that an explanation be given as to why this article was written with data taken in 2016, when there is a 6-year gap between the data collection and the analysis presented in this article.
Response: Thank you for this comment. The comment has been included.
- Results: they are presented clearly and systematically, complying with scientific requirements.
- Discussion: the discussion presented is correct, but again I suggest including more current references from the last 5 years.
Response: Thank you for the comments. The comments has been included.